# FOXO3a Mediates Homologous Recombination Repair (HRR) via Transcriptional Activation of MRE11, BRCA1, BRIP1, and RAD50

**DOI:** 10.3390/molecules27238623

**Published:** 2022-12-06

**Authors:** Gozde Inci, Madhuri Shende Warkad, Beom-Goo Kang, Na-Kyung Lee, Hong-Won Suh, Soon Sung Lim, Jaebong Kim, Sung-Chan Kim, Jae-Yong Lee

**Affiliations:** 1Department of Biochemistry, College of Medicine, Hallym University, Chuncheon 24252, Gangwon-do, Republic of Korea; 2Institute of Cell Differentiation and Aging, College of Medicine, Hallym University, Chuncheon 24252, Gangwon-do, Republic of Korea; 3Department of Food and Nutrition, The Korean Institute of Nutrition, College of Natural Science, Hallym University, Chuncheon 24252, Gangwon-do, Republic of Korea; 4Institute of Natural Medicine, College of Medicine, Hallym University, Chuncheon 24252, Gangwon-do, Republic of Korea; 5Department of Pharmacology, College of Medicine, Hallym University, Chuncheon 24252, Gangwon-do, Republic of Korea; 6FrontBio Inc. #302, Bio-2, Chuncheon BioTown, 32 Soyanggang-ro, Chuncheon 24232, Gangwon-do, Republic of Korea

**Keywords:** FOXO3a, homologous recombination repair, MRE11, BRCA1, BRIP1, RAD50

## Abstract

To test whether homologous recombination repair (HRR) depends on FOXO3a, a cellular aging model of human dermal fibroblast (HDF) and tet-on flag-h-FOXO3a transgenic mice were studied. HDF cells transfected with over-expression of wt-h-FOXO3a increased the protein levels of MRE11, BRCA1, BRIP1, and RAD50, while knock-down with siFOXO3a decreased them. The protein levels of MRE11, BRCA1, BRIP1, RAD50, and RAD51 decreased during cellular aging. Chromatin immunoprecipitation (ChIP) assay was performed on FOXO3a binding accessibility to FOXO consensus sites in human MRE11, BRCA1, BRIP1, and RAD50 promoters; the results showed FOXO3a binding decreased during cellular aging. When the tet-on flag-h-FOXO3a mice were administered doxycycline orally, the protein and mRNA levels of flag-h-FOXO3a, MRE11, BRCA1, BRIP1, and RAD50 increased in a doxycycline-dose-dependent manner. In vitro HRR assays were performed by transfection with an HR vector and I-SceI vector. The mRNA levels of the recombined GFP increased after doxycycline treatment in MEF but not in wt-MEF, and increased in young HDF comparing to old HDF, indicating that FOXO3a activates HRR. Overall, these results demonstrate that MRE11, BRCA1, BRIP1, and RAD50 are transcriptional target genes for FOXO3a, and HRR activity is increased via transcriptional activation of MRE11, BRCA1, BRIP1, and RAD50 by FOXO3a.

## 1. Introduction

One of the most deleterious types of DNA damage is the double-strand break (DSB), which can lead to growth arrest and cell death if not repaired. DSB is repaired by non-homologous end joining (NHEJ) and homologous recombination repair (HRR). NHEJ repairs double-strand breaks by direct ligation of DNA break-ends throughout the entire cell cycle [1], while HRR repairs double-strand breaks by homologous recombination mostly during the S and G2 phases [2,3]. NHEJ is an abundant repair activity for DSB in mammalian cells, but it is error-prone as end processing can lead to small deletions and translocations [4]. By comparison, HRR takes advantage of a DNA template with high sequence homology and provides high-fidelity repair for complex DNA damage including DSBs, interstrand crosslinks, and DNA gaps [5]. During HRR, 3′ single-stranded DNA (ssDNA) ends are generated by nucleolytic degradation of 5′ strands. This first step is conducted by endonucleases, including the MRN complex made up of Mre11, Rad50, and Nbs1. The ssDNA ends are then bounded by replication protein A (RPA) filaments. Finally, the complex is replaced by Rad51 in a BRCA1- and BRCA2-dependent process, and the recombination reaction is performed in a homologous DNA template. Since homologous recombination is also required for accurate chromosome segregation during the first meiotic division, defects in the HRR process are known to lead to genomic instability and enhanced cancer predisposition.

NHEJ efficiency has been reported to decrease during cellular aging of human diploid fibroblasts [6], and the NHEJ capacity of cell extracts prepared from isolated neurons also declines with age [7]. Our study found that NHEJ capacity declines with age in rat tissues. We also found that dietary restriction reverses the age-dependent decline in NHEJ [8]. NHEJ was shown to significantly decline with age in fibroblasts from the heart, lung, kidney, and skin, as well as in astrocytes from a mouse model [9]. When interchromosomal HRR levels were measured in both young and old mice, HRR efficacy declined distinctly in the pancreas, lung, and thymus, while modest declines were observed in the spleen and kidney, in old versus young mice [10]. Similar results have been reported for intrachromosomal HRR of a non-functional GFP reporter in mouse tissue [11]. Senescent fibroblasts displayed a dramatic decrease in HRR when compared to early growing fibroblasts. In this system, HRR could be stimulated in both growing and senescent fibroblasts by overexpression of SIRT6, a mono-ADP ribosyltransferase, possibly by working through activation of PARP1, an ADP-ribosylating enzyme required for commencing a variety of DNA repairs [12,13].

FOXO transcription factors belong to the O class of the forkhead transcription factor superfamily, which consists of FOXO1 (FKHR), FOXO3 (FKHRL1), FOXO4 (AFX), and FOXO6. FOXOs regulate various biological processes such as cell cycle, apoptosis, DNA repair, and reactive oxygen species (ROS) detoxification via transcriptional activation of their target genes. FOXOs bind their target genes at the consensus binding sequence of 5′-RTAAAYA-3′ [14,15,16]. FOXOs are regulated via phosphorylation by Akt [17,18], AMPK [19], JNK [20,21], MST1 [22], ERK [23], and p38 [24] as well as via acetylation/deacetylation, ubiquitination, and methylation in response to various external stimuli [25,26,27]. FOXOs appear to contribute to the maintenance of cellular and organismal homeostasis in response to stresses such as metabolic stress, oxidative stress, and growth factor deprivation [17,18,19,20,21,22,23,24,25,26,27,28,29]. Among the FOXOs, only FOXO3a is involved in detoxification of oxidative stress factors via transcriptional regulation of its target genes, MnSOD and catalase. FOXO3a is also reported to be involved in DNA repair by transcriptional activation of GADD45 [30]. FOXO3a is also known to suppress DNA double-strand-break-induced mutation [31]

To elucidate aging and FOXO3a-dependent regulation of HRR, we introduce a cellular aging model of HDF and prepared a tet-on flag-h-FOXO3a transgenic mouse model. As a detailed mechanism of how FOXO3a regulates HRR has not been clearly elucidated, in this study, we investigated the regulation mechanism. Our results demonstrate here that FOXO3a regulates HRR activity via transcriptional activation of MRE11, BRCA1, BRIP1, and RAD50 in the HDF and MEF cell system and mouse model.

## 2. Results

### 2.1. Protein Levels of MRE11, BRCA1, BRIBP1, and RAD50 Were Regulated by FOXO3a in Cellular-Aging-Dependent Manner in HDF

To test whether HRR-related genes were dependent on FOXO3a, overexpression of wt-h-FOXO3a in young HDF, knockdown with siFOXO3a in young HDF, and the cellular aging model of HDF were used. When HDF PD24 (young) cells were transfected with increasing amounts of wt-FOXO3a plasmid (0, 300, 600, 1200 ng), the protein levels of MRE11, BRCA1, BRIP1, and RAD50 increased in a FOXO3a-dose-dependent manner, but the protein levels of RAD51, BRCA2, NBS1, BARD1, and PALB2 did not, similar to that of the internal control, β-actin (Figure 1). Since HDF PD24 (young) cells were transfected with 100 nM and 300 nM of siFOXO3a, the protein levels of MRE11, BRCA1, BRIP1, and RAD50 decreased. These indicate that the regulation of MRE11, BRCA1, BRIP1, and RAD50 is FOXO3a-dependent. When the protein levels of HRR-related genes were measured during cellular aging of HDF at population doubling (PD) of 24 (“young”), 30, 36 (“middle”), 46, and 56 (“old”), the protein levels of MRE11, BRCA1, BRIP1, RAD50, and RAD51 gradually decreased, similar to the levels of FOXO3a but not those of BRCA2 and β-actin (relative) (Figure 2). These results suggest that MRE11, BRCA1, BRIP1, and RAD50 protein levels appear to be FOXO3a-dependent. However, RAD51 showed a noticeably FOXO3a-independent expression, but also decreased during cellular aging.

### 2.2. Promoter Activities of Human MRE11, BRCA1, BRIP1, and RAD50 were FOXO3a-Dependent and ChIP Assay on FOXO Consensus Sites of Human MRE11, BRCA1, BRIP1, and RAD50 Promoters Showed Cellular-Aging-Dependent Decrease in FOXO3a Binding

To test whether promoter activities of human MRE11, BRCA1, BRIP1, and RAD50 are FOXO3a-dependent, HDF cells (young, PD24) were transfected with promoter–luciferase constructs and increasing amounts of wt-h-FOXO3a, and promoter activities of these genes were measured. Promoter activities increased FOXO3a-dependently (Figure 3B,E,H,K), indicating that the transcription of these genes are FOXO3a-dependent. To test the hypothesis, FOXO3a binding to the endogenous promoters of the FOXO3a-dependentHRR genes was measured by ChIP assay. FOXO3a binding to endogenous promoters decreased during cellular aging of HDF; ChIP assays with anti-FOXO3a antibody and PCR primers were performed. The results showed that FOXO3a binding on FOXO consensus sites in human endogenous MRE11 (FOXO site #1) (Figure 3A–C), BRCA1 (FOXO site #1, #2, #3) (Figure 3D–F), BRIP1 (FOXO site #1, #2, #3, #4) (Figure 3G–I), and RAD50 promoters (FOXO site #1, #2) (Figure 3J–L) gradually decreased during cellular aging, namely at PD26 (young) and PD44 (old) (Figure 3). These results showed that the protein levels of MRE11, BRCA1, BRIP1, and RAD50 decreased during cellular aging of HDF due to decreased FOXO3a binding on FOXO consensus sites of human endogenous MRE11, BRCA1, BRIP1, and RAD50 promoters. The decreased FOXO3a binding is because of the decreased protein levels of FOXO3a during cellular aging of HDF, as shown in Figure 2.

### 2.3. The mRNA and Protein Levels of MRE11, BRCA1, BRIP1, and RAD50 Increased in a Doxycycline-Dose-Dependent Manner in Tet-On h-FOXO3a Transgenic Mice

Tet-on h-FOXO3a transgenic mice with inducible wt-h-FOXO3a were prepared from crossing of flag-h-FOXO3a-tetO transgenic mice with tet-R transgenic mice and subsequent selection of double-positive mice offspring. FOXO3a target genes to be studied were induced in the tet-on flag-h-FOXO3a transgenic mice by oral administration of doxycycline in 1% sucrose. The mRNA levels of flag-h-FOXO3a, MRE11, BRCA1, BRIP1, and RAD50 increased in a doxycycline-dose-dependent manner in mice tail-tip samples, and those for mouse FOXO3a and β-actin were not changed (Figure 4A). The protein levels of flag-h-FOXO3a, MRE11, BRCA1, BRIP1, and RAD50 increased in a doxycycline-dose-dependent manner in mice tail-tips, and those for mouse FOXO3a and β-actin were not changed (Figure 4B). The result indicates that the mRNA and protein levels of flag-h-FOXO3a, MRE11, BRCA1, BRIP1, and RAD50 were also regulated FOXO3a-dependently in vivo.

### 2.4. In Vitro HRR Activity Increased in a Doxycycline-Dose-Dependent Manner in MEF Obtained from Embryos of Tet-On h-FOXO3a Transgenic Mice and Decreased during Cellular Aging of HDF

Mouse embryonic fibroblast (MEF) cells, established from embryos of tet-on h-FOXO3a transgenic mice, and young (PD24) and old (PD46) HDF cells were used for an in vitro HRR assay. wt-MEF cells were prepared from embryos of wild-type mice (C57BL/6N) to use as a control. Before transfection of recombination plasmid, MEF cells were treated with three concentrations of doxycycline (0, 0.5, 1.5, and 4.5 μg/mL) in DMEM-10% FBS. The MEF cells and HDF cells were transfected with DR-GFP (Addgene, #26475) plasmid and I-SceI (Addgene, #26477) plasmid for the in vitro HRR assay. DR-GFP is composed of two differentially mutated GFP regions. GFP* and iGFP have mutations in the sequence of GFP and do not make functional GFP protein. Homologous recombination between GFP* and iGFP after formation of a double-strand break in the GFP* site by I-SceI produces a functional GFP. Following transfection of the MEFs with the HRR vector and I-Scel vector, the MEF cells were again treated with three concentrations of doxycycline or control (0, 0.5, 1.5, and 4.5 μg/mL) and wt-MEF cells were treated with 4.5 μg/mL of doxycycline in DMEM-10% FBS for 48 h. HDFs were also transfected with the HRR vector and I-Scel vector and incubated for 48 h. Total RNA was isolated and a real-time qPCR (quantitative PCR) was performed to assay the HRR. The results showed that the mRNA levels of the recombined GFP increased in a doxycycline-dose-dependent manner in MEF cells (from transgenic mice) but not in the wt-MEF (from wild-type mouse), and the mRNA levels of the unrecombined GFP level did not change in a doxycycline dose-dependent manner (Figure 5B); our results indicate that FOXO3a activates HRR. The mRNA levels of the recombined GFP increased doxycycline-dependently but the mRNA levels of the unrecombined GFP were unchanged (Figure 5B). The mRNA levels of both the recombined GFP and the unrecombined GFP were unchanged even though doxycycline was treated (Figure 5C). The results indicate that in vitro HRR activity is FOXO3a-dependent. The mRNA levels of the recombined GFP largely decreased (by 68%) in old (PD46) HDF cells when compared to young (PD46) HDF cells (Figure 5B). The result suggests that cellular-aging-dependent decrease of FOXO3a levels cause a decrease in HRR.

## 3. Discussion

Regulation of homologous recombination repair (HRR) by FOXO transcription factors has not been reported. Here, we showed that FOXO3a activates HRR through transcriptional activation of its target genes, MRE11, BRIP1, RAD50, and BRCA1, using a cellular aging model of HDF and a model of tet-on flag-h-FOXO3a inducible transgenic mice (Figure 6). As a DSB is one of the most deleterious DNA lesions, the cell repairs DSBs by two pathways, NHEJ and HRR. When HRR is deficient, DSB repair leads to enhanced dependence on alternative pathways including NHEJ, alternative end joining, and single-strand annealing [32,33]. However, these pathways repair DSBs without a homologous DNA template, and are error prone, often producing small deletions and translocations around the DSB [4]. As such, HRR deficiency has been closely related to increases in cellular DNA mutation and increased incidence of cancers, particularly for breast and ovarian cancers [34,35,36,37]. As homologous recombination (HR) is also required for chromosome segregation during meiotic division, defects in the HR process also lead to genomic instability and enhanced cancer predisposition.

On the regulation of HRR, FOXM1 has been implicated in the activation of HRR. FOXM1 has been reported to upregulate BRIP1 [38] and NBS1 [39], and FOXM1 activation upregulates RAD51 and BRCA1 in idiopathic pulmonary fibroblasts to activate HRR [40], with FOXM1 being a transcription factor that plays an important role in proliferation, cell cycle control, DNA repair, tumorigenesis, cancer progression, and tumor growth [41,42]. FOXO3a is reported to transcriptionally regulate GADD45 to mediate DNA repair [30], and is known to suppress DNA double-strand-break-induced mutation [43]. FOXO3a has also been reported to be upregulated at the transcriptional and translational level to activate DSB repair in bleomycin-treated MEF and HDF [31]. Therefore, FOXO3a may be involved in regulation of HRR activity. As a transcription factor, FOXM1 appears to activate the expression of target genes involved in HRR such as BRCA1/2, RAD51, BRIP1, and NBS1. The forkhead transcription factors, FOXM1 and FOXO3a, however, have been reported to suppress each other. Overexpression of FOXM1 resulted in downregulation of FOXO3a, and overexpression of FOXO3a induced downregulation of FOXM1 [44,45]. In this context, FOXO3a overexpression is expected to suppress the levels of RAD51, BRIP1, and BRCA1/2 those upregulated by FOXM1 overexpression. However, our results show that this is not true in the system of our study. FOXO3a overexpression was seen to induce transcriptional activation of MRE11, BRCA1, BRIP1, and RAD50 via increased interaction of FOXO3a with FOXO consensus sites of promoters of these genes (Figure 2, Figure 3, Figure 4 and Figure 5), and FOXO3a overexpression induced activation of HRR (Figure 6). In our system, FOXM1 overexpression caused upregulation of MRE11, BRIP1, BRCA1, RAD50, and RAD51 and downregulation of FOXO3a (data not shown). FOXM1 overexpression and FOXO3a overexpression were similar in upregulating MRE11, BRCA1, BRIP1, and RAD50. The difference was that FOXM1 overexpression induced RAD51 and BRCA2, but FOXO3a overexpression did not. Therefore, as both FOXM1 and FOXO3a are decreased during cellular aging, overexpression of either FOXM1 or FOXO3a activates HRR.

The details of regulation of HRR by FOXM1 and FOXO3a remain to be solved. FOXO3a has been reported to interact with ATM to promote phosphorylation of ATM [45]. FOXM1 has also been shown to be upregulated following exposure to radiation by an undescribed mechanism [39]. Further characterization of the detailed regulation of FOXM1/FOXO3a by a DNA damage signal (ATM or γ-H2AX) and the detailed regulation of HRR by FOXM1/FOXO3a will provide a better understanding of the regulation of HRR.

Cellular aging in HDF cells decreased the levels of FOXO3a, MRE11, BRCA1, BRIP1, RAD50, and RAD51 (Figure 2). HRR activity also declines distinctly in many tissues of old mice [10], and senescent fibroblasts display a dramatic decrease in HRR when compared to early growing fibroblasts [12,13]. FOXO3a levels have been shown to decrease in senescent fibroblasts [46] with a decreased binding to FOXO consensus sites of target genes in aged *Drosophila* [47]. HRR decreases in aged cells appear to be due to decreased levels and binding of FOXO3a to the target promoters involved in mediating HRR. We also observed a decrease in the levels of FOXM1 in aged cells (data not shown). The reduced levels of FOXM1 in aged cells also appears to contribute to decreased HRR in aged cells. On the separate or joint contribution of FOXO3a and FOXM1 to the reduced levels of HRR in aged cells, a detailed study is necessary for a better understanding. Additional contributions to reduced HRR in senescent fibroblasts may be via changes in the levels of SIRT6, as HRR was stimulated by overexpression of SIRT6, possibly through the activation of PARP1 [12,13]. Thus, in addition to FOXO3a/FOXM1, contributions of SIRT6 and PARP1 to HRR also merit a detailed study. We only showed aging-related regulation of HRR by FOXO3a in cellular aging of HDF cells, but not in body aging. Because cellular aging does not exactly represent body aging, regulation of HRR by FOXO3a should be further verified in body aging. In summary, in this study, key transcriptional targets of FOXO3a for HRR in the context of the aging models and the tet-on flag-h-FOXO3a inducible transgenic mouse were identified, pointing to a level of regulation by FOXO3a in modulating a key DNA repair mechanism.

## 4. Materials and Methods

### 4.1. Cell Culture and Transfections

Human dermal fibroblast (HDF) cells were obtained from the Dermatology Laboratory of Seoul National University Medical School (Seoul, Republic of Korea). HDF cells were transfected by Lipofectamine 3000 (Invitrogen, Waltham, MA, USA) according to the manufacturer’s instructions. The reaction was performed with 30 µL of Lipofectamine 3000 added to 0.5 mL of DMEM (Dulbecco’s modified Eagle’s medium, Thermo Fisher Scientific, Waltham, MA, USA), and 300 ng, 600 ng, or 1200 ng of wt-h-FOXO3a plasmid (Addgene, #8360) (Watertown, MA, USA) or HDF PD24 (young) cells were transfected with siFOXO3a (100 nM or 300 nM) (Cell Signaling Technology, Danvers, MA, USA) dissolved in 2 µL of Lipofectamine 3000 and 0.5 mL DMEM, respectively. The two solutions were mixed and added to 10 mL culture of HDF in DMEM in a 10 cm dish and incubated at 37 °C for 6 h. The HDF cells were then incubated at 37 °C for overnight in DMEM and supplemented with 10% fetal bovine serum (FBS) (Thermo Fisher) also containing an antibiotics mix of penicillin and streptomycin (Thermo Fisher). The HDF cells were subsequently used in Western blotting, ChIP and luciferase reporter assays. Transfection efficiency was determined by pCMV-beta-Gal-transfected HDF cells, according to the manufacturer’s instructions (Takara Bio, Kusatsu, Shiga, Japan).

### 4.2. Primary Mouse Embryonic Fibroblasts

All animal protocols were approved by the Hallym University Institutional Animal Care and Use Committee (IACUC) (approval H20180119). Primary mouse embryonic fibroblasts (MEFs) were obtained from the embryos of pregnant tet-on flag-h-FOXO3a transgenic mice, and wt-MEFs were obtained from the embryos of pregnant wild-type mice (C57BL/6N). The embryos (embryonic day 12.5–14.5) of pregnant transgenic mice were taken out after incision surgery and rinsed in sterile cold PBS on ice to remove blood. These were then decapitated and their visible internal organs were removed, followed by mincing with sterile blade into small fragments as fine as possible and cultured in 10 cm dishes containing DMEM (Thermo Fisher), 10% FBS (Thermo Fisher), and penicillin/streptomycin (Thermo Fisher) at 37 °C in a humidified atmosphere containing 5% CO_2_. MEF cells were treated with control and three concentrations of doxycycline (0, 0.5, 1.5, and 4.5 μg/mL) and wt-MEF cells were treated with doxycycline (0, 4.5 μg/mL) DMEM-10% FBS for 24 h before transfection. Lipofectamine 3000 (L3000) (Invitrogen) was used for transfection of MEF cells according to the manufacturer’s instructions.

### 4.3. Western Blot Analysis

HDF cells or tail-tip samples of tet-on flag-h-FOXO3a transgenic mice were lysed in lysis buffer (50 mM Tris-HCl, pH 7.4; 150 mM NaCl; 1 mM EDTA; 0.25% sodium deoxycholate; 1% NP-40; and supplemented with a protease inhibitor cocktail) (Sigma-Aldrich, St. Louis, MO, USA). A total of 25 micrograms of protein extract from the HDF cells or 40 μg of protein extract from the tail-tip samples were separated on an SDS–polyacrylamide gel and transferred to an ImmunoBlot PVDF membrane. The membranes were incubated with primary antibodies, washed, and then incubated with horseradish peroxidase-conjugated secondary antibodies. After washing, the resulting protein bands were visualized by ECL (Amersham, Little Chalfont, UK). The Western probing antibodies for FOXO3a, MRE11, BRCA1, BRCA2, BRIP1, RAD50, and RAD51 were purchased from Santa Cruz Biotechnology (Dallas, TX, USA); antibodies for PALB2, NBS1, and BARD1 were purchased from Proteintech (Rosemont, USA); Anti-Flag and anti-beta actin antibodies were from Cell Signaling Technology (Danvers, MA, USA).

### 4.4. Plasmid Constructs

The promoter region of the human MRE11 gene was amplified from human genomic DNA by PCR using the following primer sets, and the PCR products were inserted into NheI and SacI sites of pGL3-basic vector (pGL3-MRE11). The primers were MRE11 (forward), 5′-GCGTAGCTAGCTATCAACTGCATAGATAAGTCTTGCAAACATTAC-3′, and MRE11 (reverse), 5′-GCTACGAGCTCGAATCTATCTGAACCTCCTTCACCAGTAAGGT-3′. Similarly, the promoter region of the human BRCA1 gene was amplified and inserted into *Nhe*I and *Sac*I sites of pGL3-basic vector (pGL3-BRCA1) using BRCA1 (forward), 5′-GCG TA GCTAGCGAAGGATCATGAGCCTAGGAGTTCAAGACA-3′, and BRCA1 (reverse), 5′-GCTACGAGCTCGGCTTATTACGTCACAGTAATTGCTG TACCA-3′, primers. In the same fashion, the promoter region of the BRIP1 gene was amplified and cloned into *Nhe*I and *Sac*I sites of pGL3-Basic (pGL3-BRIP1). The primers were BRIP1 (forward), 5′-GCGTAGCTAGCACGGTTCAAGGGACTGTATTCGAGGTCCA-3′, and BRIP1 (reverse), 5′-GCTACGAGCTCGAGGCGGAAGGTTGTCGCCACTCCAGCA-3′. Similar to the above, the promoter region of RAD50 gene was amplified and cloned into *Nhe*I and *Sac*I sites of pGL3-Basic (pGL3-RAD50) with primers RAD50 (forward), 5′-GCGTAGCTAGCAATGGAACCGGTTACCTTGGCATGTCCA-3′, and RAD50 (reverse), 5′-GCTACGAGCTCGGTGGCTCACGCCTGTAATCCCAGCACT-3′.

### 4.5. Promoter Reporter Assay

HDF PD26 cells were transfected with the human promoter reporter plasmids (pGL3-MRE11, pGL3-BRCA1, pGL3-BRIP1, and pGL3-RAD50) and an increasing amount of wt-h-FOXO3a using Lipofectamine 3000 transfection reagent (Invitrogen) according to the manufacturer’s instructions. The cell extracts were then prepared by incubating cells in cell lysis buffer (Promega, Madison, WI, USA) and the luciferase activity of the cell extracts was measured with a luminometer (GloMAX 20/20, Promega) using a luciferase assay reagent (Promega).

### 4.6. ChIP Assay

The ChIP assay was performed on young HDF (PD26) and old HDF (PD44) cells. Cells were crosslinked with 1% formaldehyde at room temperature for 10 min and washed twice with 1 mL PBS containing 0.5 mM PMSF. The cells were then lysed in 50 µL ice-cold lysis buffer (50 mM HEPES-KOH, pH 7.5; 140 mM NaCl; 1 mM EDTA; pH 8.0; 1% Triton X-100; 0.1% sodium deoxycholate; and 0.1% SDS) containing 0.5 mM PMSF and were incubated on ice for 30 min. To obtain the average DNA fragments sizes of 300–1000 bp, the cell lysates were sonicated using a sonicator (Vibra Cell, Sonics & Materials, Newtown, CT, USA). The cell lysates were then diluted in 100 µL dilution buffer (50 mM Tris-HCl, pH 8.0; 150 mM NaCl; 2 mM EDTA; 1% NP-40; 0.5% sodium deoxycholate; 0.1% SDS; and protease inhibitors) plus 0.5 mM PMSF. Immunoprecipitation was carried out by mixing single-stranded salmon sperm DNA (1 µg), protein A/G beads (20 µL), BSA (2 µg), and 2 μg of anti-FOXO3a antibody (isotype anti-IgG antibody as a negative control) containing 25 µg of DNA in 100 µL RIPA buffer (50 mM Tris-HCl, pH 8.0; 150 mM NaCl; 2 mM EDTA; 1% NP-40; 0.5% sodium deoxycholate; 0.1% SDS; and protease inhibitors). The reaction was incubated overnight with slow rotation at 4 °C. The mix was then centrifuged to pellet the A/G beads, which were then washed twice by the wash buffer. The beads were then in 20 µL of elution buffer per sample to obtain isolated DNA for the subsequent PCR reaction. PCR amplification of FOXO3a-binding consensus sites in the human MRE11 promoter was performed with two sets of primers: MRE11 site 1 (forward), 5′-CAATTCATTAAACACTTAACA-3′, and MRE11 site 1 (reverse), 5′-TGTATCTGGATCTGTAAACTC-3′. PCR amplification of the FOXO3a-binding consensus sequences of the human BRCA1 promoter was carried out using the following two sets of primers: BRCA1 site 1 (forward), 5′-ATGGCGTGAACCTGGGAGGTGG-3′; BRCA1 site 1 (reverse), 5′-GGTGAACTTGGTTCTTGGTAGAA-3′; BRCA1 site 3 (forward), 5′GGGCGACAGAGCAAGACTCCAT-3′; and BRCA1 site 3 (reverse), 5′-GGTGAACTTGGTTCTTGGTAGAA-3′. PCR amplification of FOXO3a-binding consensus sites in the human BRIP1 promoter was performed with the following two sets of primers: BRIP1 site 1 (forward), 5′-GAAATTTAACTTTAGCCTTTTCAC-3′; BRIP1 site 1 (reverse), 5′-CTATCACATGTGATACTATGCACT-3′; BRIP1 site 4 (forward), 5′-GATGTGTGCAGCTATTTTGAATAT-3′; and BRIP1 site 4 (reverse), 5′-GTTCCCTTCAGGTTAACTCTAGTG-3′. PCR amplification of FOXO3a-binding consensus sites in the human RAD50 promoter was performed using the following two sets of primers: RAD50 site 1 (forward), 5′-GTGACAGAGTGAGACTCCATC-3′; RAD50 site 1 (reverse), 5′-GATGAGTGTGAAGTGATACCTC-3′; RAD50 site 2 (forward), 5′-CATGGATGAACCTTGAGGA-3′; and RAD50 site 2 (reverse), 5′- ACTCACGTACAACCGTCAC-3′. The PCR reactions were performed with the following parameters: initial 5 min at 95 °C, followed by 35 cycles of 10 s at 94 °C, 30 s at 57–63 °C, and 30 s at 72 °C, and a final 10 min incubation at 72 °C. The PCR products were separated for analysis by electrophoresis in a 1.5% agarose gel.

### 4.7. RNA Extraction and Quantitative Real-Time RT-PCR (RT-qPCR)

Total RNA was purified from tail-tip samples of tet-on flag-h-FOXO3a transgenic mice using an RNeasy Mini Kit (Qiagen, Germany, Hilden). Total RNA at 0.5 μg was reverse-transcribed into cDNA using oligo (dT) and the RT-PCR System SuperScript II Reverse Transcriptase kit (Thermo Fisher) in accordance with the manufacturer’s instructions. RT-qPCR was then carried out with TOPreal qPCR 2X PreMIX reagent with SYBER Green/high ROX (Enzynomics, DaeJeon, Republic of Korea) per the manufacturer’s instructions. The following primers were used: MRE11 forward, ATGAGCCCCACAGATCCACTT; MRE11 reverse, CCTTCTCCACCGACATTGAC; BRCA1 forward, CAAGAACCGGTTTCCAAAGA; BRCA1 reverse, TGAAATATTCATGCCAGAGGTC; BRIP1 forward, GCAACCACCAGTCCATATC; BRIP1 reverse, AGCCAATCATCGTACGAGG; RAD50 forward, GCGTGCGAAGTTTTGGGAT; RAD50 reverse, TTGTTTCTTTCGGCTCTCCA; β-actin forward, CATTGCTGACAGGATGCAGAAGG; β-actin reverse, TGCTGGAAGGT GGACAGTGAGG; FOXO3a forward, TACGAGTGGATGGTGCGCTG; FOXO3a reverse, AGGTTGTGCCGGATGGAGTTC; flag-h-FOXO3a forward, GACTACAAGG ACGATGACGATA; and flag-h-FOXO3a reverse, GTCCAGCTCCACTTCGAGC. The conditions for PCR were initial denaturation at 95 °C for 15 min and 40 cycles of PCR (denaturation at 95 °C for 10 s, annealing at 60 °C for 15 s, and elongation at 72 °C for 30 s).

### 4.8. Development of the Tet-On Flag-h-FOXO3a Transgenic Mice

Ph-FOXO3a-tetO (5.17 kb) was constructed by insertion of flag-human FOXO3a cDNA between SV40 promoter-tet-operator and SV40 poly (A) signal in a vector. Ph-FOXO3a-tetO was then linearized by digestion with restriction enzymes and separated by agarose gel electrophoresis. The DNA was purified using a gel extraction kit (Qiagen). The purified DNA was microinjected at a concentration of 10 μg/mL in PBS into fertilized C57/BL6 mouse eggs (Macrogen, Seoul, Republic of Korea). Transgenic animals were identified by PCR of genomic DNA obtained from tail-tip samples. The sense and antisense primers were 5′-GCGGAGCGAGGAACTGAG-3′ and 5′-CCGCCCTGGGA ATGATAG-3′, respectively. pLenti6-tet-R was constructed in a vector to include the CMV promoter/β-globin intron/tet-R/poly(A). Linearized pLenti6-tet-R was microinjected at a concentration of 10 μg/mL in PBS into fertilized C57/BL6 mouse eggs (Macrogen). Transgenic animals were identified by PCR of genomic DNA obtained from the tail-tip sampling. The sense and antisense primers were 5′-GCGGAGCGAGGAACTGAG-3′ and 5′-CCGCCCTGGGA ATGATAG-3′, respectively. Two transgenic mice were mated and double-positive mice (flag-h-FOXO3a-tetO and tet-R) were obtained, as confirmed by PCR of tail-tip genomic DNA. All animal experimental protocols were approved by the Laboratory Animal Committee of Hallym University (IACUC) (approval number Hallym 2019-17). Twelve double-positive tet-on flag-h-FOXO3a transgenic mice (12 weeks old) were used for testing flag-h-FOXO3a-dependent gene expression in the tail tissue samples. Carrier-only or solution containing doxycycline at 0.2 mL (0, 0.1, 1, 10 mg/mL 1% sucrose) was orally administered to transgenic mice once a day for 2 days. Cell extracts were prepared from the tail-tip samples on day 3, and their proteins were separated and analyzed by Western blotting. Anti-flag, anti-FOXO3a, anti-RAD50, anti-BRCA1, anti-BRIP1, anti-MRE11, anti-flag, and anti-β-actin antibodies were used for protein identification.

### 4.9. In Vitro Homologous Recombination Repair (HRR) Assay

In vitro homologous recombination assay was performed as described previously [44]. Briefly, MEF cells were established from embryos of tet-on flag-h-FOXO3a mice, and wt-MEF cells were prepared from C57BL/6N wild mice. MEF (from tet-on flag-h-FOXO3a transgenic mice) cells were treated with three concentrations of doxycycline (0, 0.5, 1.5, and 4.5 μg/mL); wt-MEF cells were treated 4.5 μg/mL doxycycline. Lipofectamine 3000 (L3000) (Invitrogen) was used for transfection of MEF cells as described in the section of cell culture and transfection. DR-GFP (Addgene, #26475) plasmid is composed of two differentially mutated GFP regions. The downstream GFP gene and the upstream GFP sequence contained the inactive I-SceI site within the GFP sequence before the homologous recombination repair. MEF cells were transfected with 2 µg DR-GFP plasmid and 2 µg I-SceI (Addgene, #26477) plasmid as described in the cell culture and transfection section. I-SceI expression vector was used to induce a double-strand break at a genomic I-SceI site. After transfection with the plasmids, MEF cells were treated with three concentrations of doxycycline or control (0, 0.5, 1.5, and 4.5 μg/mL) in DMEM-10% FBS for 48 h. Total RNA was prepared from MEF cells as described in the section for RNA extraction and quantitative real-time RT-qPCR. The cDNAs from unrecombined and recombined GFP mRNAs were synthesized from 1 µg total RNA in 20 µL by quantitative qPCR. Two sets of primers were prepared for use in RT-qPCR. The un-recombination (unrec) primer sequence was localized on the I-SceI site present only upstream of GFP; the second primer set was the recombinant (rec) primer sequence present downstream of GFP. The forward primer sequences for unrec was 5′-GCTAGGGATAACAGGGTAAT-3′; for rec, it was 5′-GAGGGCGAGGGCGATGCC-3′; for the reverse primer, it was 5′-TGCACGCTGCCGTCCTCG-3′. The conditions were 95 °C for 15 min, followed by 40 cycles of 95 °C for 10 s, 60 °C for 15 s, and 72 °C for 35 s. The PCR products were resolved and visualized in a 1.5% agarose gel.

### 4.10. Statistical Analysis

Statistical analysis was carried out by Student’s t-test using the GraphPad Prism Version 4.0 software for Windows (GraphPad Software, San Diego, CA, USA). *p*-values less than 0.05 were considered to indicate statistical significance. All values are expressed as mean ± S.E.M.

## 5. Conclusions

The regulation mechanism of HRR activity by FOXO3a was addressed in HDF cells during cellular aging and in a FOXO3a-inducible transgenic mouse model system. MRE11, BRCA1, BRIP1, and RAD50 among nine HRR factors turned out to be regulated cellular-aging-dependently and transcriptionally by FOXO3a in a cellular aging model of human dermal fibroblast (HDF). Furthermore, FOXO3a upregulated MRE11, BRCA1, BRIP1, and RAD50 at transcriptional and translational levels in tet-on flag-h-FOXO3a inducible transgenic mice. Likewise, HRR activity was shown to be regulated FOXO3a-dependently in an in vitro DR-GFP reporter HRR assay using an MEF (mouse embryonic fibroblast) obtained from the tet-on flag-h-FOXO3a inducible transgenic mice. In conclusion, MRE11, BRCA1, BRIP1, and RAD50 are transcriptional target genes of FOXO3a, and FOXO3a contributes to homologous recombination repair activity in MEF cells and mouse model systems.

## Figures and Tables

**Figure 1 molecules-27-08623-f001:**
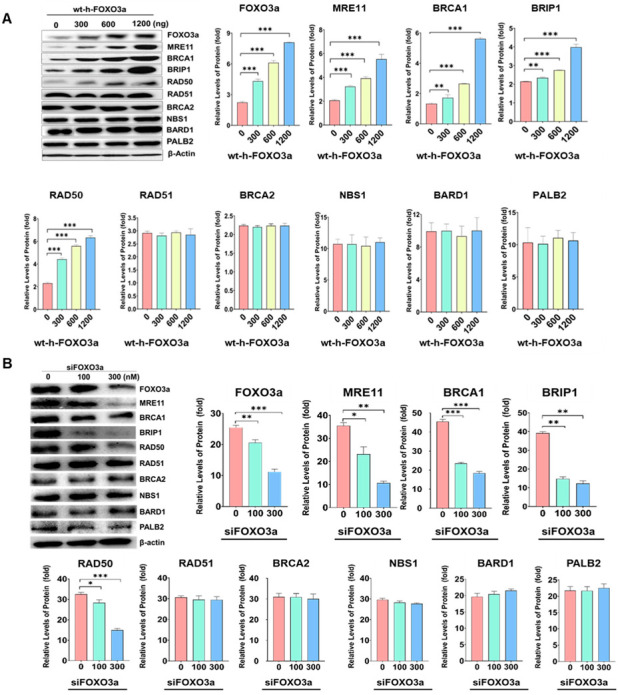
(**A**) The protein levels of MRE11, BRCA1, BRIP1, RAD50, RAD51, FOXO3a, NBS1, BARD1, PALB2, and β-actin were measured by Western blotting in HDF cells transfected with increasing amounts of wt-h-FOXO3a plasmid. HDF cells of PD24 (young) were transfected with 0, 300, 600, and 1200 ng of wt-h-FOXO3a using Lipofectamine 3000. The HDF cells were then cultured for 24 h. Transfection efficiency was determined by pCMV-beta-Gal-transfected HDF. Relative band intensities of MRE11, BRCA1, BRIP1, RAD50, RAD51, NBS1, BARD1, PALB2, and FOXO3a versus β-actin were measured by densitometry and the data were plotted as histograms. (**B**) The protein levels of MRE11, BRCA1, BRIP1, and RAD50 were measured in siFOXO3a-transfected HDF PD24 (young) cells by Western blotting. Standard deviations as error bars were obtained from three different experiments. Statistical significance is indicated as * *p* < 0.05, ** *p* < 0.01, and *** *p* < 0.001.

**Figure 2 molecules-27-08623-f002:**
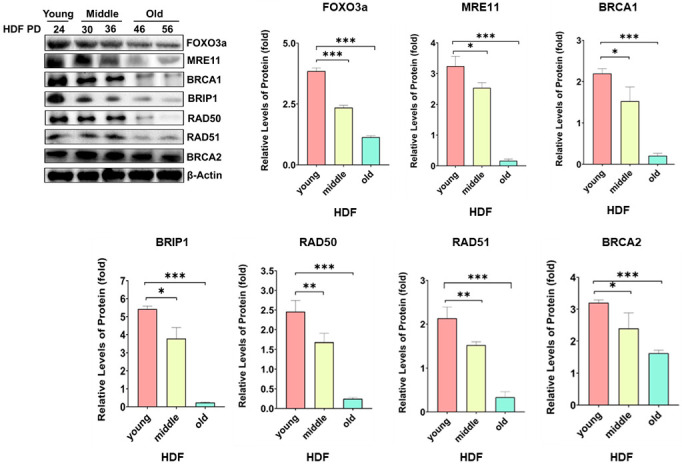
The protein levels of MRE11, BRCA1, BRIP1, RAD50, RAD51, BRCA2, FOXO3a, and β-actin were measured by Western blotting. Cell extracts were prepared from HDF cells at PD24 (“young”), PD30, PD36 (“middle”), PD46, and PD56 (“old”). Relative band intensities of MRE11, BRCA1, BRIP1, RAD50, RAD51, BRCA2, and FOXO3a versus β-actin were measured by densitometry and the data were plotted as histograms. Standard deviations as error bars were obtained from three different experiments. Statistical significance is indicated as * *p* < 0.05, ** *p* < 0.01, and *** *p* < 0.001.

**Figure 3 molecules-27-08623-f003:**
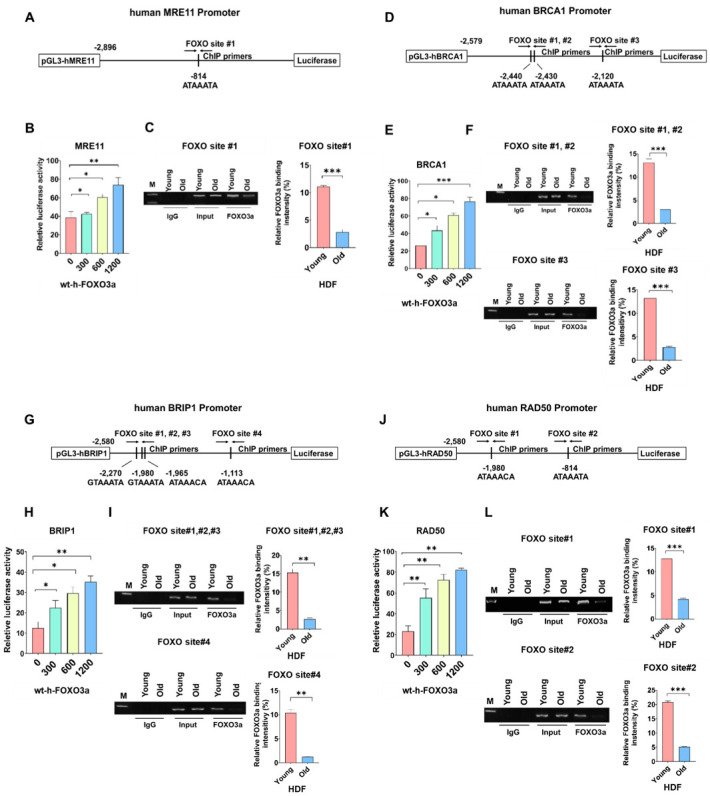
MRE11, BRCA1, BRIP1, and RAD50 promoter activities were measured in young HDF cells transfected with increasing levels of wt-h-FOXO3a plasmid. FOXO3a binding to FOXO consensus sites of each promoter was measured by ChIP assay during cellular aging of HDF. The schematic diagrams of (**A**) human MRE11 promoter–luciferase (pGL3-hMRE11), (**D**) human BRCA1 promoter–luciferase (pGL3-hBRCA1), (**G**) human BRIP1 promoter–luciferase (pGL3-hBRIP1), and (**J**) human RAD50 promoter–luciferase (pGL3-hRAD50) constructs depicting the FOXO-binding consensus sites and ChIP-primer-binding locations were shown. HDF cells were transfected with 0, 300, 600, and 1200 ng of wt-h-FOXO3a and (**B**) pGL3-hMRE11, (**E**) pGL3-hBRCA1, (**H**) pGL3-BRIP1, and (**K**) pGL3-RAD50 plasmids using Lipofectamine 3000. HDF cells were then incubated in DMEM for 24 h. The luciferase activity of the cell extracts was measured with a luminometer (GloMAX 20/20, Promega). Cell lysates were prepared from HDF cells of PD26 (“young”) and PD44 (“old”). Cell lysates were sonicated on ice to obtain sheared average DNA fragments of 300 bp to 1000 bp. Immunoprecipitation was carried out as described in the Materials and Methods section. Isolated genomic DNA from immunoprecipitation was used in the ChIP-PCR reaction. ChIP-PCR amplification of FOXO3a-binding consensus sites in (**C**) the human MRE11 promoter, (**F**) the human BRCA1 promoter, (**I**) the human BRIP1 promoter, and (**L**) the human RAD50 promoter was performed as described in the Materials and Methods section. The PCR products were separated by electrophoresis in 1.5% agarose gel. Standard deviations as error bars were obtained from three different experiments. Statistical significance is indicated as * *p* < 0.05, ** *p* < 0.01, and *** *p* < 0.001.

**Figure 4 molecules-27-08623-f004:**
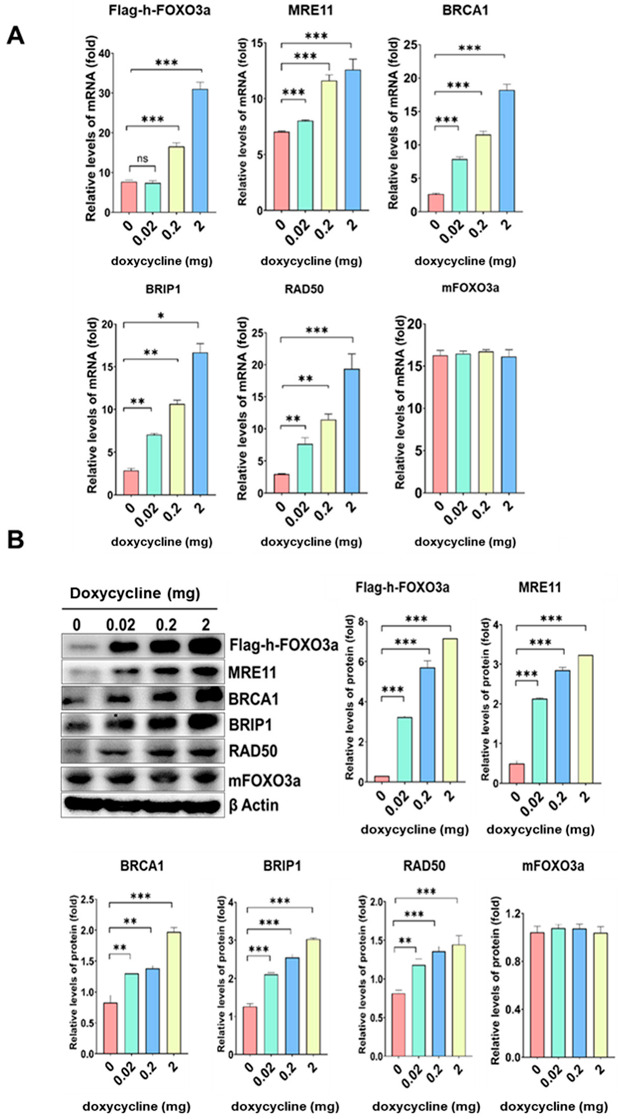
FOXO3a-dependent upregulation of MRE11, BRCA1, BRIP1, and RAD50 mRNA and protein expression in tet-on flag-h-FOXO3a transgenic mice. The tet-on flag-h-FOXO3a transgenic mice were used to test flag-h-FOXO3a-dependent gene expression in the tail-tips of the mice. Oral administration of 0.2 mL of doxycycline (0, 0.02, 0.2, and 2 mg in 1% sucrose) solution was performed on the transgenic mice once a day for 2 days. (**A**) RNAs were prepared from the tail-tips on day 3. RNA was analyzed by quantitative real-time RT-PCR as described in the Materials and Methods section. Relative amounts of flag-h-FOXO3a, MRE11, BRCA1, BRIP1, RAD50, and mouse FOXO3a mRNAs versus β-actin mRNA were measured and the data were plotted as histograms. (**B**) Cell extracts were prepared from the tail-tip samples on day 3. Protein levels were analyzed by Western blotting using anti-flag, anti-MRE11, anti-BRCA1, anti-BRIP1, anti-RAD50, anti-FOXO3a, and anti-β-actin antibodies. Relative band intensities of flag-h-FOXO3a, MRE11, BRCA1, BRIP1, RAD50, and mFOXO3a versus β-actin were measured by densitometry and the data were plotted as histograms. Standard deviations as error bars were obtained from three different experiments. Statistical significance is indicated as * *p* < 0.05, ** *p* < 0.01, and *** *p* < 0.001.

**Figure 5 molecules-27-08623-f005:**
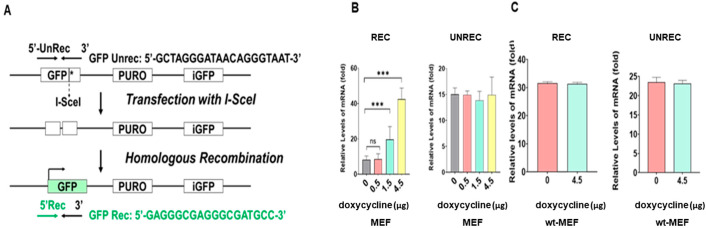
In vitro HRR activity was measured by transfection of MEF and wt-MEF with DR-GFP (Addgene, #26475) plasmid and I-SceI (Addgene, #26477) plasmid. Following transfection of the MEFs with the HRR vector and I-Scel vector, the MEF cells were again treated with three concentrations of doxycycline or control (0, 0.5, 1.5, and 4.5 μg/mL) and wt-MEF cells were treated with 4.5 μg/mL of doxycycline in DMEM-10% FBS for 48 h. HDFs were also transfected with the HRR vector and I-Scel vector and incubated for 48 h. Old HDF cells were transfected with wt-h-FOXO3a for 24 h to induce FOXO3a before HRR assay to test the restoration of HRR activity. Total RNAs were isolated and RT-qPCRs for recombined and unrecombined GFP were performed to measure HRR activity. (**A**) Schematic diagram depicting the in vitro HRR assay. (**B**) RT-qPCR for total RNA was performed to detect recombined and unrecombined GFP in MEF (from tet-on h-FOXO3a transgenic mice) and (**C**) wt-MEF (from wild-type mice). Statistical significance is indicated as *** *p* < 0.001, ns: not significant.

**Figure 6 molecules-27-08623-f006:**
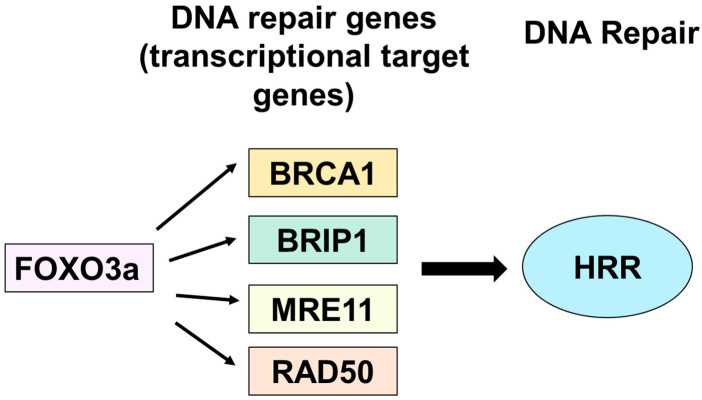
A schematic diagram showing up-regulation of HRR activity via transcriptional activation of target HRR genes by FOXO3a.

## Data Availability

Not applicable.

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
