# Peer review of "FOXO3a Mediates Homologous Recombination Repair (HRR) via Transcriptional Activation of MRE11, BRCA1, BRIP1, and RAD50"

_molecules, 2022, doi:10.3390/molecules27238623_

Round 1
Reviewer 1 Report (Previous Reviewer 1)
Revised manuscript is suitable for publication
Author Response
Thank you for your comments.
Reviewer 2 Report (Previous Reviewer 3)
In the article "FOXO3 mediates homologous recombination repair (HRR) via transcriptional activation of MRE11, BRCA1, BRIP1 and RAD50.
The paper is clearly written and easy to understand.
This manuscript is potentially interesting given the importance of understanding how double strand break repair is regulated.
The authors do over come some of the previous comment by adding cell cycle data, which is needed.
Yet, this manuscript still falls short for quality of data. Again, all western blot experiments can come under question. The authors have been asked repeated to add loading controls for each experiment western blot.
Worse yet, in figure 1B, the authors have two actin western blots which are clearly the same where one is just lightened and cropped. This is highly inappropriate.
Also, similar to previous review of the manuscript, there is several instances where data presented in the manuscript is not consistent with the original data.
In figure 1A, BRCA1/2, Rad50 and NBS1 are appear to be different western blots.
In figure 1B, MRE11, BRIP1, RAD51, and RAD50 are appear to be different western blots.
In Figure 1 FOXO3a, MRE11 and BRCA1 appear to be different western blots.
This lack of rigor, detail and quality of western blot data is extremely detrimental to the entire manuscript, as it forms the foundation for the conclusions made.
Author Response
We appreciate your proper comments. Your comments were very helpful for us to realize general etiquette in writing and revising manuscript.
- Yet, this this manuscript still falls short for quality of data. Again, all western blot experiments can come under question. The authors have been asked repeated to add loading controls for each experiment western blot.
Worse yet, in figure 1B, the authors have two actin western blots which are clearly the same where one is just lightened and cropped. This is highly inappropriate.
Our Answer:
Although we repeated at least 3 times for all western blot analyses, we did not add all repeats in our “Supplementary Data File”. We are very sorry for this. It is apparently our mistake and we are responsible for this. We now added all the repeats of the original western blot analysis results to “ThSupplementary Data File”. It was our mistake to add two same copies of actin western blots. We feel really sorry about this. However, we like to say that it is just a mistake but not a manipulation. We added now three original blots in “Supplementary Data File”.
- Also, similar to previous review of the manuscript, there is several instances where data presented in the manuscript is not consistent with the original data.
In figure 1A, BRCA1/2, Rad50 and NBS1 are appear to be different western blots.
In figure 1B, MRE11, BRIP1, RAD51, and RAD50 are appear to be different western blots.
In Figure 1 FOXO3a, MRE11 and BRCA1 appear to be different western blots.
This lack of rigor, detail and quality of western blot data is extremely detrimental to the entire manuscript, as it forms the foundation for the conclusions made.
Our Answer:
We changed the previous blots with the better blots among three repeats when we revised the manuscripts. We are again very sorry for our unusual change. We showed all repeated blots this time in the “Supplementary Data File”. Your comments are proper and we are responsible for getting this blame. We will not try again for unusual changes to the original figures in the manuscript without having the reviewer’s comment.
Please see the attachment file.

Reviewer 3 Report (New Reviewer)
While it has been established that DNA repair declines with aging, the mechanism poorly understood. Here, the authors attempt to understand whether FOX3a play a role in the decline of HRR during aging. The first part of the study looks at whether FOXO3a increases the expression of HRR factors and whether FOXO3a declines with age and how that affects the expression of several HRR factors that they have selected to measure. This part of the study is well designed. The authors convincing show that FOXO3a expression correlates with MRE11, BRCA1, BRIP1, RAD51 and RAD52 and that these factors decrease in aging cells. They confirm these results in their mouse model. However, they have not gone far enough to show that the age-related decrease is FOXO3a-dependent. Gene expression of many factors is known to decrease with cellular age. An experiment, such as overexpressing FOXO3a in the old HDF cells to rescue expression of the selected HRR factors, would provide more concrete evidence that the decline in the aging model is FOXO3a-dependent.
The luciferase and ChIP data do lend support for their model, but it was unclear to me why they used the plasmid-based luciferase construct for the ChIP analysis. Analysis of the endogenous promoters would have been more convincing. Nevertheless, these data do support the idea that FOXO3a directly regulates these HRR factors. The authors then use the HDR plasmid assay to assess HRR, the major goal of the study. However, I was perplexed as to why they chose to measure the mRNA levels of GFP versus looking at GFP positive cells, as the assay is setup to do. Is there are a reason for this? They did include a rescue experiment in the HDF model, but it only partially rescued suggesting that other factors contribute. This should be discussed.
Based on this single assay, the authors then conclude that FOXO3a expression activates HRR. While I agree that FOXO3a appears to regulate the expression of the specific HRR genes, at present that data supporting the idea that this leads to HRR defects is minimal and preliminary. Additional studies would require to support their conclusion. For example, evidence that these factors do not localize to DSBs in aging cells and/or that breaks remain unrepaired or lead to increased genome instability, which is then rescued by overexpression of FOXO3a, or similar types of experiments should be performed. Moreover, a previous study that the authors cite (White et al. Aging Cell 2020) showed that FOXO3a expression is increased following induced DSB formation so it would also be insightful to test how FOXO3a and HRR factors expression is affected after genome wide DSB induction in old versus young cells. Alternatively, the authors could modify their conclusions to only include their findings that FOXO3a increases expression of these HRR factors and this could potentially affect HRR, based on their HDR results. This finding alone is significant.
Author Response
We appreciate your excellent comments. Your comments were very helpful to figure out the details of the relationship between HRR and FOXO3a
1. An experiment, such as overexpressing FOXO3a in the old HDF cells to rescue expression of the selected HRR factors, would provide more concrete evidence that the decline in the aging model is FOXO3a-dependent.
Our Answer: We have already analyzed HRR factors by western blotting. The result was in the supplementary data file and we decided to add the western result in Figure 5E. The protein levels of MRE11, BRIP1, BRCA1, and RAD50 were elevated by overexpression of FOXO3a in old HDF cells. We added a sentence to describe the result in Result section (the line between 7-9 on page 12) and, we added cell extracts prepared from FOXO3a overexpressed old HDF cells, and proteins were analyzed by western blot analysis. (E) in Figure legend of Figure 5 (on page 13, line between 6-7) in the manuscript.
- The luciferase and ChIP data do lend support for their model, but it was unclear to me why they used the plasmid-based luciferase construct for the ChIP analysis. Analysis of the endogenous promoters would have been more convincing.
Our Answer: The plasmid based–promoter-luciferase constructs were used only for promoter assay in Figure 3B, 3E, 3H, and 3K. We used the endogenous promoters for the ChIP analysis in Figure 3C, 3F, 3I and 3L. We added the word “endogeneous” to get rid of confusion in Result section (page 7, line 8,9,11 and 15) and a word “genomic” in Figure legend of Figure 3C, 3F, 3I and 3L (page 9, line 12).
- The authors then use the HDR plasmid assay to assess HRR, the major goal of the study. However, I was perplexed as to why they chose to measure the mRNA levels of GFP versus looking at GFP positive cells, as the assay is setup to do. Is there are a reason for this?
Our Answer: We also have attempted to count GFP positive cells under the fluorescence microscope. However, it was not sensitive and convenient in our hands when compared to measuring the mRNA levels of GFP by RT-qPCR. Therefore, we decide to use this way.
- They did include a rescue experiment in the HDF model, but it only partially rescued suggesting that other factors contribute. This should be discussed.
Our Answer: Old HDF cells showed less than 3% of transfection efficiency by using Lipofectamine 3000 although young HDF showed higher than 5% of transfection efficiency. Therefore, we expected much less rescue of HRR by wt-FOXO3a in old HDF when considering low transfection efficiency. However, the result showed much higher than expected and the rescued HRR activity was enough high to see the difference. We added sentences describing the situation of transfection efficiency in Result section (Page 12, line between 6-9) and Discussion section (Page 14, last paragraph last sentence, page 15 line between 1-2). We changed the sentence in Result section with “The decreased in vitro HRR activity in old HDF was restored largely by transfection of wt-h-FOXO3a (Figure 5D) although transfection efficiency of wt-FOXO3a was very low in old HDF cells.” And the sentences in Discussion section with “Although activation of HRR by FOXO3a overexpression was not same as HRR of young HDF, it was enough high to show FOXO3a-depedent rescue in old HDF cells. Because transfection efficiency was less than 3% in old HDF cells, this high rescue of HRR was not expected by transfection of wt-FOXO3a in old HDF cells”.
- Based on this single assay, the authors then conclude that FOXO3a expression activates HRR. While I agree that FOXO3a appears to regulate the expression of the specific HRR genes, at present that data supporting the idea that this leads to HRR defects is minimal and preliminary. Additional studies would require to support their conclusion. For example, evidence that these factors do not localize to DSBs in aging cells and/or that breaks remain unrepaired or lead to increased genome instability, which is then rescued by overexpression of FOXO3a, or similar types of experiments should be performed. Moreover, a previous study that the authors cite (White et al. Aging Cell 2020) showed that FOXO3a expression is increased following induced DSB formation so it would also be insightful to test how FOXO3a and HRR factors expression is affected after genome wide DSB induction in old versus young cells. Alternatively, the authors could modify their conclusions to only include their findings that FOXO3a increases expression of these HRR factors and this could potentially affect HRR, based on their HDR results. This finding alone is significant.
Our Answer: We agree that our data supporting the idea that this leads to HRR defects is minimal and preliminary. As you suggested, HRR factors will not be co-localized on DSB in old cells (can be analyzed by immunofluorescence) and overexpression of FOXO3a in old cells will show increase in co-localization of HRR factors to DSB and decrease of genome instability. We think it is a good way to prove involvement of FOXO3a in repairing DSB via HRR. However, we already have the unpublished preliminary result of FOXO3a involvement in NHEJ (non-homologous end-joining). FOXO3a up-regulates transcriptionally one of NHEJ factor and FOXO3a overexpression activates NHEJ activity. FOXO3a overexpression can remove DSB in old cells through both HRR and NHEJ. NHEJ may be the major activity for the removal of DSB as expected. Therefore, we cannot figure out what percentage of DSB is solved by HRR. Increase in co-localization of HRR factor to DSB in old cells by FOXO3a overexpression may not be a good way to show how FOXO3a overexpression rescue specifically HRR in old cells since HRR and NHEJ are overlapped for DSB. So we think even though HRR assay does not look like the best, it appears to be the only way to measure FOXO3a involvement solely in HRR activity.
Because cellular aging does not represent body aging, cellular aging dependent decrease of HRR may not be true in body aging. We are working on 3 year aging study of tet-on-h-FOXO3a transgenic mice. We will prepare MEF cells obtained from young (2 weeks old) transgenic mice and old mice (1, 2, 3 year old) treated with and without doxycycline for short or long time and these MEF cells were compared for HRR assay. This will show whether HRR factors and HRR activity are down-regulated and FOXO3a overexpression may also rescue HRR activity in MEF cells obtained from bodily aged mice. We added several sentences describing possible difference in HRR regulation by FOXO3a between cellular and body aging in Discussion section (Page 15, line between 23-25). “We only showed aging-related regulation of HRR by FOXO3a in cellular aging of HDF cells but not in body aging. Because cellular aging does not represent exactly body aging, regulation of HRR by FOXO3a should be further verified in body aging.”
Please see attached file.

Round 2
Reviewer 2 Report (Previous Reviewer 3)
This version of paper is highly improved from previous version. There are no concerns with manuscript
Author Response
Thank you for your comments.
Reviewer 3 Report (New Reviewer)
While the authors have addressed many of my concerns, some have not been adequately addressed. Most importantly, the authors did not add new data to support their conclusions/model, or soften their conclusions to better represent their current data. For example, the title implies that FOXO3a is required to mediate HRR but this a correlation at best right now. I am confused by their explanation that they have preliminary data that NHEJ is also stimulated by FOXO3a so they may not be able to measure the percentage of HRR. If their model is that HRR is improved with FOXO3a expression, then this should be measurable in cells. If not, it is unclear why this study is relevant. HRR and NHEJ can also be separated by looking at different phases of the cell cycle. As mentioned previously, the authors can just modify their conclusions to better fit their result, unless they add additional data to support their model. I am also confused by their explanation of the rescue experiment (point #4). They state that the transfection efficient is only 3-5%. in these cells. If correct, how do they achieve the ~50% rescue shown in Fig 5D? Based on these points, I do not support publication of the study in its current form.
Author Response
Our Answer:
I appreciate your excellent comments and suggestions. Your comments helped us to realize what is important to make our results convincing. As you gave us a comment, we also think that our result was not enough to make our conclusion by our results about the rescue of HRR activity by FOXO3a overexpression in old HDF cells. I am also sorry to make you confused by bringing up the regulation of NHEJ by FOXO3a and transfection efficiency. We decided to remove Figure 5D which showed poor rescue of HRR activity by FOXO3a in old HDF cells and to change our conclusion in our manuscript as you suggested to modify. The following sentences were removed in the Result section. “The decreased in vitro HRR activity in old HDF was restored largely by transfection of wt-h-FOXO3a (Figure 5D) although transfection efficiency of wt-FOXO3a was very low in old HDF cells. The protein levels of MRE11, BRCA1, BRIP1, and RAD50 were elevated in old HDF by transfection of wt-h-FOXO3a as expected (Figure 5E).” (line 7-10, on page 12) and the Figure legend 5D and 5E of Figure 5 were removed. “(D) young and old HDF cells. wt-h-FOXO3a was overexpressed in old HDF cells to restore HRR activity. Total RNA was isolated and an RT-qPCR for total RNA was performed to test HRR as described in the methods section. The PCR products were separated by electrophoresis in 1.5% agarose gel. Standard deviations as error bars were obtained from three different experiments. (E) Cell extracts were prepared from old HDF cells elevated with doxycycline and proteins were analyzed western blot analysis.” (line 10-14, on page 13). We removed a sentence in the result section (lines 17-20, on pages 11-12). “HDF PD 46 old cells were transfected with wt-FOXO3a (600 ng), in DMEM-10% FBS for 24 hr before transfection.” We also removed a word in the result section (line 16, on page 11). “HDF cells.”
We also removed sentences as follows in the Discussion section (lines 19-21 on page 14). Although activation of HRR by FOXO3a overexpression was not sane as HRR of young HDF, it was enough high to show FOXO3a-dependent rescue in old HDF cells. Because transfection efficiency was less than 3% in old HDF cells, this high rescue of HRR was not expected by transfection of wt-FOXO3a in old HDF cells.
We removed apart in the conclusion section (line 13, on page 22) “human dermal fibroblast (HDF).” We removed words in the conclusion section (line 15, on page 22). “HDF cells.”

This manuscript is a resubmission of an earlier submission. The following is a list of the peer review reports and author responses from that submission.
Round 1
Reviewer 1 Report
The manuscript by Inci et al. describes several experimental approaches to test the hypothesis that a number of genes known to encode important members of the cellular homologous recombinational repair (HRR) machinery are subject to regulation by the FOXO3a transcription factor. The authors show that overexpression of FOXO3a in human dermal fibroblasts as well as in transgenic murine embryonic fibroblasts resulted in increased expression of the MRE11, BRCA1 BRIP and RAD50 genes. The authors report that overexpression of FOX3a in the murine cell system increased the frequency of homologous recombination using a DNA double-strand break repair/recombination plasmid system. The authors also described changes in expression levels of the FOXO3a and HRR genes in a cellular aging model using human dermal fibroblasts.
Strengths of the manuscript include the clarity with which it was written (however see below) and the excellent quality of the data, which indeed strongly support the authors’ central hypothesis. Additional strengths include the variety of experimental approaches that were used including ChIP, luciferase-based promoter reporter systems, RT-qPCR, western blotting, and the aforementioned plasmid-based DNA double-strand break-repair induced HRR assay. In all instances the data are clearly presented, with appropriate statistical analysis included, and the experimental approaches used are well described.
The biggest defect with the manuscript is the ‘conclusions’ section, which appears to be a nearly identical copy of the discussion. The redundancy is, of course, unacceptable. Also, there are a number of instances where the citations in the two otherwise identical copies don’t agree. It seemed to me that the ‘discussion’ section correctly cited the references and the ‘conclusions’ section is where the mis-citations occurred. Also, references 45 and 48 are identical.
Reviewer 2 Report
Gozde et al present a study describing the FOXO3a mediates homologous recombination repair (HRR) via transcriptional activation of MRE11, BRCA1, BRIP1 and RAD50 using human dermal fibroblasts as model system. Although interesting, the authors didnot explain why they chose this model system in the first place and should include more information in the introduction section. Also, the authors should discuss the opposite effects of FOXO3a in their study as compared to study published by R.R.White et al, where they show FOXO3a suppresses homologous repair using the same model system of mouse embryonic fibroblasts and human dermal fibroblasts. How do the authors claim their study is different from what is already published. Please explain.
The study can be improved and following revisions will be needed to re-consider it for publication.
1. siRNA/shRNA studies against FOXO3a should be performed to see opposite effects as shown by the authors using overexpression system.
2. Figure 1 should mention what stage HDF were used (young, old, middle).
3. FOXO3a should be knocked in young HDF to confirm FOXO3a direct role in regulating HRR proteins.
4. Figure showing DR-GFP HR assay should also include results from NHEJ-reporter assay to confirm FOXO3a only regulates HRR and doesn’t affect NHEJ capacity of the cells. Similarly, western blot analysis should be done include NHEJ proteins.
5. Results from WT-MEF in figure 6b are missing.
6. Rescue experiments in old HDF should be included to see HRR is rescued by FOXO3a expression in these cells.
Manuscript needs language editing.
Reviewer 3 Report
In the manuscript "FOXO3a mediates homologous recombination repair (HRR) via transcriptional activation of MRE11, BRCA1, BRIP1 and RAD50", the authors attempt to determine how FOXO3a mediates expression of HRR proteins in young vs old cells.
This is a potentially interesting manuscript.
The manuscript is clearly written and easy to follow.
This manuscript is also over a an important topic.
Yet, enthusiasm for paper is diminished for the following reason.:
This manuscript is not properly controlled, bringing into question the results presented.
While a loading control is provided for western blots, it is done a separate gel making comparisons indirect.
It appears that the authors do not take into account cell cycle distribution between young and old cells and how this may influence the results.
For ChIP assay, the authors use exongenous plasmids. While they perform luciferase assays to determine that cells where transfected, it appears that they overlook that there may be a difference in how many plasmids are introduced into young cells vs old cells.
Specific comments:
Figures 1/2: Figures are not properly controlled. Loading control must be examined for each individual HRR protein examined.
Figure 3: The amount of plasmids introduced in young and old cells is not examined. The use of the endogenous HRR promoter would enhance the data.
Figures 4/5 can be combined.
Figure 6: There is no cell cycle analysis performed. If more old cells are in G1 than young cells, then the difference observed may be due to cell cycle rather than ability to perform HRR.
Round 2
Reviewer 2 Report
Authors have provided a point by point clarification in the revised manuscript and have conducted appropriate experiments as suggested. The revised manuscript is accepted.
Reviewer 3 Report
This is a reresubmission of a previously reviewed manuscript. This authors of this manuscript are attempting to describe a role for FOXO3a in homology driven DNA repair by modulating expression multiple repair regulators.
This manuscript is consistent with other publications indicating a role for FOXO3a in DNA repair.
Yet, the current manuscript is while improved still suffers in quality and rigor in multiple areas:
1: There is no data presented on differences in cell-cycle distribution between young and old cells. In response, author claim to have performed studies but do not present them.
2: Western blots on figure 1 and 2 are not properly controlled. There is a single beta actin blot given for multiple different target proteins. This actin blot is not even show together with a target protein, but on a blot all by itself.
3: There doe not appear to be any original blots for figure 1B
4: All original blots are not labeled. Thus making it hard to compare data presented to originals.
5: Figure 5 is not properly controlled. This type of experiment is typically performed with co-transfection of a plasmid expression RFP as a transfection control.
6: In response, the author claim that over-expression of FOXO3a increases HR repair efficency without changing cell-cycle distribution. The author makes no attempt to explain how this occurs. Does over-expression FOXO3a activate HR in G1 cells? This would be important if true and there should at least be some discussion of how this is occuring.
7: There are also several differences between western blots presented in manuscript as compared to original blots given:
In Figure 1, the FOXO3a, MRE11, RAD50, RAD51, BRCA2 and beta actin blots appear to be differnet
In Figure 2, the beta actin, RAD51, FOXO3a, RAD50, MRE11 and BRIP1 are diffenent
In Figure 4, the BRIP1, BRCA1, FOXO3a and over-expressed FOXO3a blots are different.
In some cases this may be due to the original being in wrong orientation, yet without any labels on blots this is left to the reader to imagine. In other cases there appears to be totally different western blots presented from originals.